# Physically and Chemically Crosslinked Hyaluronic Acid-Based Hydrogels Differentially Promote Axonal Outgrowth from Neural Tissue Cultures

**DOI:** 10.3390/biomimetics9030140

**Published:** 2024-02-25

**Authors:** Andrej Bajic, Brittmarie Andersson, Alexander Ossinger, Shima Tavakoli, Oommen P. Varghese, Nikos Schizas

**Affiliations:** 1Department of Surgical Sciences, Section of Orthopedics, The OrthoLab, Uppsala University, 75185 Uppsala, Sweden; andrej.bajic@uu.se (A.B.); brittmarie.andersson@uu.se (B.A.); alexander.ossinger@uu.se (A.O.); 2Translational Chemical Biology Laboratory, Division of Macromolecular Chemistry, Department of Chemistry-Ångstrom Laboratory, Uppsala University, 75237 Uppsala, Sweden; shima.tavakoli@kemi.uu.se (S.T.); oommen.varghese@kemi.uu.se (O.P.V.)

**Keywords:** axonal outgrowth, neuroprotection, HA-based hydrogel, spinal cord slice culture, dorsal root ganglion culture

## Abstract

Our aim was to investigate axonal outgrowth from different tissue models on soft biomaterials based on hyaluronic acid (HA). We hypothesized that HA-based hydrogels differentially promote axonal outgrowth from different neural tissues. Spinal cord sliced cultures (SCSCs) and dorsal root ganglion cultures (DRGCs) were maintained on a collagen gel, a physically crosslinked HA-based hydrogel (Healon 5^®^) and a novel chemically crosslinked HA-based hydrogel, with or without the presence of neurotrophic factors (NF). Time-lapse microscopy was performed after two, five and eight days, where axonal outgrowth was assessed by automated image analysis. Neuroprotection was investigated by PCR. Outgrowth was observed in all groups; however, in the collagen group, it was scarce. At the middle timepoint, outgrowth from SCSCs was superior in both HA-based groups compared to collagen, regardless of the presence of NF. In DRGCs, the outgrowth in Healon 5^®^ with NF was significantly higher compared to the rest of the groups. PCR revealed upregulation of NeuN gene expression in the HA-based groups compared to controls after excitotoxic injury. The differences in neurite outgrowth from the two different tissue models suggest that axons differentially respond to the two types of biomaterials.

## 1. Introduction

Limited functional recovery after neuronal injury is often associated with poor axonal regeneration. Moreover, massive neuron death that follows the injury also contributes to overall neurological deficit [1,2,3,4]. Attempts to promote axonal regeneration and neuroprotection both after spinal cord injuries and peripheral nerve injuries have been made by using numerous biomaterials [5,6]. The environment of the extracellular matrix (ECM) plays a major role in sustaining intercellular interaction [5]. It has been shown in previous studies that spinal cord slice cultures (SCSCs) maintained on a hydrazone crosslinked hyaluronic acid (HA)-based hydrogel were superior in terms of neuronal survival compared to SCSCs maintained on a collagen gel [7]. However, the potential of HA-based hydrogels both crosslinked and non-crosslinked has not been investigated in terms of axonal outgrowth.

Neuroprotective and neurotrophic factors such as brain-derived neurotrophic factor (BDNF), glial-derived neurotrophic factor (GDNF) and neurotrophin (NT)-3 have also shown positive effects in terms of neuroprotection and axonal regeneration. Neurotrophins promote neuronal survival, stimulate axonal growth, and play a key role in the construction of the normal synaptic network [8]. There has been an investigation of the effects of BDNF, GDNF, and NT-3 when used alone or in combination to induce the outgrowth of spiral ganglion cells. It was found that GDNF alone was as effective as the combination of all three in enhancing neurite outgrowth [9,10]. There is great research interest in the combination of the best-performing gels with neurotrophic and neuroprotective factors.

Biocompatible ECM-based hydrogels are a mainstay in ongoing research for numerous areas including axonal regeneration. The ECM is the microenvironment that surrounds cells and is critical for functional and biochemical support [5,6]. In the central nervous system (CNS), ECM composition is vastly different than that of peripheral tissues. Peripheral ECM is composed primarily of collagens, laminins, and fibronectins, whereas in the spinal cord, it is composed of a loose meshwork of HA [11]. It is therefore rational to use biocompatible hydrogels based on HA components as a promising substrate for axonal regeneration. In addition, HA-based hydrogels behave as three-dimensional hydrophilic networks capable of absorbing large volumes of water or biological agents. This renders them ideal candidates for drug delivery vectors and carriers/matrices for cells in tissue engineering because the active ingredients can reach the target cells/tissues with higher bioavailability due to immune system evasion or increased membrane permissibility [12].

It is apparent that HA-based hydrogels have shown great potential for promoting axonal regeneration; however, additional research is still needed to investigate the most optimal composition in order to design reliable nerve substitutes. In many cases, after neuronal injury, the biomaterial must be shaped in situ with the crosslinking reaction taking place in a delicate environment of cells under stress. Additionally, the biomaterial formulation must not be associated with the release of toxic side products. Therefore, we have developed a variety of hydrogels with low degrees of modification and mild chemical reactions that allow the retention of native biological properties. Before planning in vivo studies to assess the efficacy of the hydrogels over a long period, we set out to investigate and compare the regenerative and neuroprotective effects of two HA-based hydrogels (physically and chemically crosslinked) and a collagen gel in two in vitro models of neuroprotection and axonal regeneration. The hydrogel obtained from covalent crosslinking employed thiazolidine linkages between HA-chains, distinguishing it from the other two biomaterials that were commercially procured with physical entanglement between chains. We have recently demonstrated the unique properties of thiazolidine-based HA hydrogels, showcasing their ability to enhance cell proliferation and exhibit prolonged stability compared to dynamically covalent crosslinked HA hydrogels [13]. The thiazolidine-linked HA hydrogel has a rapid gelation rate, making it particularly well suited to efficient cell encapsulation applications.

Our aim was to promote axonal outgrowth in two different tissue models of axonal regeneration through biocompatible biomaterials and neuroprotective and neurotrophic factors. Furthermore, we aimed to highlight the neuroprotective properties of HA-based hydrogels on spinal cord tissue. We hypothesized that the biomaterials based on HA differentially promote axonal sprouting depending on the type of tissue culture as compared to biomaterials based on collagen.

## 2. Materials and Methods

### 2.1. Summary of Axonal Regeneration Experiment

Neural tissue cultures were obtained from postnatal mouse spinal cord and dorsal root ganglion. SCSCs and dorsal root ganglion cultures (DRGCs) were maintained in three different biomaterials, creating a 3D matrix, i.e., collagen gel, physically crosslinked HA-based hydrogel (Healon 5^®^, Abbott, Sweden, also referred to as “non-crosslinked” for the purpose of the study) and a chemically crosslinked HA-based hydrogel, also referred to as “crosslinked” for the purpose of the study. All experiments were conducted under aseptic conditions using sterile instruments and after the approval of the local ethics committee (DNR 5.8.18-16672/2019).

Tissue cultures were maintained in the presence of culture medium (CM) or neurotrophine medium (NM), resulting in six groups of at least 10 and up to 35 cultures for SCSCs and DRGCs (Collagen CM, Collagen NM, Healon 5^®^ CM, Healon 5^®^ NM, Crosslinked CM, Crosslinked NM). CM consisted of 50% MEM (Statens Veterinarmedicinska Anstalt [SVA], Uppsala, Sweden), 25% Hank’s balanced salt solution (HBSS; Gibco Life Technologies, Stockholm, Sweden), 25% normal horse serum (NHS; Gibco Life Technologies), 2% glutamine powder (Sigma, Stockholm, Sweden), 1 μg/mL insulin (Sigma), 2.4 mg/mL glucose (Sigma), 0.1 mg/mL streptomycin (SVA), 100 U/mL penicillin (SVA) and 0.8 μg/mL vitamin C (Sigma). For NM, the factors BDNF, GDNF, and NT3 were added in CM. The tissue cultures were maintained at a pH = 7.4 and incubated at 35 °C and in a 5% CO_2_-enriched atmosphere.

Time-lapse photography was performed after two, five and eight days in vitro, where axonal outgrowth was investigated by light microscopy and automated image analysis using a novel upgraded approach called NeuriteSegmentation manual body [14]. To ensure that the investigated structures were actually axonal projections, a number of cultures were immunohistochemically double-stained for neurofilament-L (NF-L) and growth associated protein-43 (GAP-43).

### 2.2. Summary of Hydrogel-Induced Neuroprotection Experiment

After culture preparation, SCSCs were transferred in culture wells containing CM and divided into five groups (*n* = 10–12 per group), as shown in Table 1. Three out of five groups were excitotoxically injured by adding a 4 μmol/mL NMDA in CM. After a 4 h period, the CM was replaced with fresh CM and the cultures were transferred in culture wells containing HA-based hydrogels or CN, as shown in Table 1. A total of five groups were created and maintained for an additional 4 h period (total of 8 h) and then shock-frozen in liquid nitrogen and maintained at −70 °C until further processing. Neuronal survival in SCSCs was evaluated by gene analysis of the pan-neural marker NeuN after NMDA-induced excitotoxic injury.

### 2.3. Preparation and Maintenance of Dorsal Root Ganglion Cultures

Postnatal p10 C57BL/6 mice were euthanatized by decapitation and the skin of the ventral part of the animal was surgically removed. The visceral organs were subsequently removed and the vertebral column of the animal exposed. The animal was moved to a plastic container filled with phosphate-buffered saline (PBS), which was placed under a light microscope, where a microscopic dissection was performed. Microsurgical instruments were used to remove the anterior part of the vertebral column to expose the spinal cord. DRGs were located, carefully extracted and inserted into culture wells. After an incubation period of two, five and eight days, images were captured using a Leica DFC 3000G camera mounted on a phase-contrast microscope (Leica DMi8 Incubator).

### 2.4. Preparation and Maintenance of Spinal Cord Sliced Cultures

Spinal cords were extracted from postnatal mice (p10). After decapitation, the skin above the sacral and lumbar regions was surgically removed, exposing the lamina. The lumbar spine was detached from the sacrum and ice-cold preparation medium (minimal essential medium [MEM] containing 1% glutamine powder, pH = 7.35) was injected into the caudal spinal canal using a 23-gauge cannula to flush out the spinal cord through the cervical spine. The spinal cord was then immediately transferred to a tube containing ice-cold preparation medium, placed on a filter paper and chopped using a tissue chopper (McIlwain Tissue Chopper; Mickle Laboratory Engineering, Surrey, UK) into 500 μm slices. The slices were then transferred into culture wells.

### 2.5. Preparation of Biomaterials (Collagen Gel and HA-Based Hydrogels)

Collagen gel (rat-tail collagen) was mixed with sodium bicarbonate and minimal essential medium and incubated for 2 h at 37 °C in order to allow gel formation. Then, 900 μL of rat-tail collagen 3.68 mg/mL (BD Biosciences) was mixed with 110 μL MEM and 4.4 μL sodium bicarbonate 7.5%, resulting in a final collagen concentration of about 3.4 mg/mL. A total of 500 μL was coated in each PET membrane and incubated at 37 °C for 2 h. Cultures were then placed in 6-well culture plates containing 1 mL culture medium. The cultures were subsequently incubated at 35 °C and the culture medium was changed every other day.

For the fabrication of chemically crosslinked HA-based hydrogels, aldehyde-modified HA (HA-CHO) and cysteine-modified HA (HA-CYS) with a ≈10% degree of modification were dissolved separately in PBS for 1–2 h. The pH of the dissolved components was precisely adjusted to approximately 7.4 using 0.1 mL NaOH. Subsequently, the two solutions were combined in a 1:1 ratio, initiating the formation of the thiazolidine-crosslinked hydrogel through the reaction between the cysteine (CYS) and aldehyde (CHO) functionalities [13].

The physically crosslinked HA-based hydrogel, Healon 5^®^, was obtained from Abbott (Uppsala, Sweden). It was extracted from rooster combs and dissolved in a physiological buffer solution at a concentration of 23 mg/mL. It had an osmolarity and pH similar to the aqueous humor of the eye, and the product was originally intended for use during cataract surgery in the human eye.

### 2.6. Tissue Fixation

SCSCs and DRGCs were fixated using Zamboni’s fixative, a mixture of paraformaldehyde and picric acid: 4% paraformaldehyde (VWR, Stockholm, Sweden) in 0.2 mol/L Sorensen phosphate buffer, pH = 7.3, containing 0.2% picric acid (Histolab, Gothenburg, Sweden) for 24 h and subsequently washed with a 20% sucrose solution (pH = 7.3, Sigma-Aldrich, Darmstadt, Germany). Sucrose washing was performed on a daily basis until the yellow staining of the fixative disappeared. Unsectioned cultures were maintained in sucrose solution until staining.

### 2.7. Quantitative Analysis

The open-source macro NeuriteSegmentation manual body was implemented as a macro for Image J version 1.51f and used for image analysis of growing axons [14]. Analysis in NeuriteSegmentation manual body was performed as described previously [14], and datasheets of the area occupied by the explant, the area occupied by the axons, and the distance for each pixel in the segmented neurites to the explant (all measured in pixels) were saved and transferred to Excel sheets (Microsoft, Albuquerque, NM, USA). The parameters of examination were as follows: (i) axonal density, i.e., the total number of pixels of growing axons; (ii) mean distance, i.e., the average distance in pixels from the explant; and (iii) max distance, i.e., the maximal axon growth from the explant.

### 2.8. Gene Expression Analysis

The frozen SCSCs were mechanically homogenized (TissueLyser II, Qiagen, Tokyo, Japan), and total RNA was isolated from each sample using TRIzol Reagent (Invitrogen, Waltham, MA, USA) according to manufacturer’s protocol. Total RNA yield was measured by a NanoDrop spectrophotometer (ND-1000), and equal amounts of RNA from each sample were used for reverse transcription (High-Capacity RNA-to-cDNA Kit, Applied Biosystems, Waltham, MA, USA). The resulting cDNA was used to measure expression of IL-1β and NeuN via a real-time quantitative polymerase chain reaction (RT-qPCR) (7500 Fast RT PCR System, Applied Biosystems), using Actin-β as a housekeeping gene. The gene expression assays used in our experiment are shown in Table 2.

### 2.9. Statistical Analysis

Statistical analysis was conducted using SPSS V.28 software. A one-way ANOVA followed by a post hoc Dunnet’s control was performed, wherein the samples were normally distributed. Non-parametric Kruskal–Wallis and Mann–Whitney U-tests were performed, wherein the samples were not normally distributed. The Wilcoxon test for dependent groups was performed between different timepoints during the time-lapse experiment. The level of significance was set to *p* ≤ 0.05 and is indicated with * throughout the analysis. The results are presented as means ± standard error of the mean (SEM).

## 3. Results

Axonal growth was observed in all the groups in both tissue culture models. Time-lapse photography was performed with brightfield microscopy at an early timepoint (day 2), middle timepoint (day 5) and late timepoint (day 8) on tissue cultures in all biomaterials except collagen-based gel because of very scarce axonal growth (Figure 1A,D). Tissue cultures maintained in the collagen-based biomaterials were assessed at the middle timepoint (day 5).

### 3.1. Axonal Outgrowth from SCSCs Maintained in HA-Based Hydrogels and Collagen

At two days in vitro, the crosslinked HA-based hydrogel was superior compared to the non-crosslinked gel in terms of axonal density in SCSCs when neurotrophic factors (NM) were added in the medium (Figure 2B). At the middle timepoint, both crosslinked and non-crosslinked HA-based hydrogels were superior compared to collagen gel in terms of axonal density and average axon length, regardless of the medium used (Figure 2A–D). In the absence of trophic factors (CM), we observed longer axons in the non-crosslinked Healon 5^®^ hydrogel; however, there were no differences in terms of average axon length and axonal density between the gels (Figure 2C). A migration of cells was observed in the non-crosslinked hydrogel in the presence of neurotrophic factors, which was also evident in DRGCs (Figure 1). At the late timepoint, SCSCs showed clear superiority in axonal growth in non-crosslinked Healon 5^®^ hydrogel compared to the crosslinked gel in terms of axonal density, average and maximal axon length, regardless of the medium used (Figure 2A–F). In SCSCs, axons continued to grow after 8 days when maintained in non-crosslinked Healon 5^®^ hydrogel, while they remained stable when maintained in a crosslinked hydrogel (Figure 2A–F).

### 3.2. HA-Based Hydrogels Compared to Collagen Gel in Terms of Outgrowth from DRGCs; the Effects of Outgrowth in Non-Crosslinked HA-Based Hydrogel Compared to Crosslinked Hydrogel in the Presence of Neurotrophic Factors

At the early timepoint in DRGCs, we observed a significant increase in axonal density in Healon5^®^ compared to the crosslinked hydrogel when neurotrophic factors were added (Figure 3B). Other parameters including mean distance and max distance remained stable regardless of which medium was used. At the middle timepoint and in the absence of trophic factors, the axonal outgrowth from DRGCs in crosslinked HA-based hydrogel was superior compared to the other two biomaterials in terms of average and maximal axon length (Figure 3C,E). In the presence of NM, we observed a clear superiority of both HA-based hydrogels compared to collagen gel in terms of axonal density and average and maximal axon length (Figure 3B,D,F).

In the crosslinked hydrogel, the outgrowth in DRGCs remained stable between 5 and 8 days. At the late timepoint and in the absence of neurotrophic factors, DRGCs grew superior in the crosslinked HA-based hydrogel compared to non-crosslinked Healon5^®^ hydrogel in terms of average axon length (Figure 3C).

However, non-crosslinked Healon5^®^ hydrogel was superior in terms of axonal density (Figure 3A). When neurotrophic factors were added, the outgrowth in DRGCs on non-crosslinked Healon5^®^ hydrogel reached a level equal to that in crosslinked gel in terms of average axon length, and it remained superior in terms of axonal density (Figure 3B,D).

When maintained in non-crosslinked hydrogel in the absence of neurotrophic factors, DRGCs grew in average length but not in terms of density. In the presence of neurotrophic factors, the DRGCs continued to grow significantly in terms of density and average length even after 8 days.

### 3.3. Effects of HA-Based Hydrogels on Excitotoxically Damaged SCSCs

We observed a clear downregulation of NeuN gene expression in the NMDA-injured group compared to NMDA-injured cultures maintained in the presence of HA-based hydrogel regardless of if it was crosslinked or non-crosslinked. Specifically, the groups maintained in HA-based hydrogel after NMDA excitotoxic injury showed an almost 1.5-fold upregulation of NeuN gene expression (Figure 4A). At the same time, a clear downregulation of IL-beta gene expression in the NMDA-injured group was observed compared to the rest of the groups (Figure 4B).

## 4. Discussion

The literature surrounding HA-based hydrogels aims to investigate and characterise the neuroprotective effect of the gels and neuroprotective substances used [15,16,17]. However, there are a lack of of investigations into the process of combining HA-based hydrogel with neurotrophic factors through the use of an in vitro model. In furthering this outgrowth model by describing the outgrowth observed from the DRGCs and SCSCs, we may reach a deeper understanding of what occurs in vivo, thus allowing for the development of implants that could be used in neural regeneration and neuroprotection.

### 4.1. HA-Based Hydrogels Are Superior to Collagen Gel in Terms of Axonal Sprouting

We have shown in our study that HA-based hydrogels, both crosslinked and not, promote axonal outgrowth of mouse-derived SCSCs and DRGCs to a significantly greater extent compared to collagen-based hydrogel. This can partially be explained by the fact that the extracellular matrix (ECM) in the neural structures mostly consists of hyaluronic acid rather than collagen. Cell–matrix interactions are fundamental to many developmental, homeostatic, immune and pathologic processes. HA, a critical component of the extracellular matrix that regulates normal structural integrity and development, also regulates tissue responses during injury, repair, and regeneration. Neural ECM is radically different from those of other tissues. The interstitial ECMs are mainly composed of a loose meshwork of hyaluronan, sulfated proteoglycans, and tenascins. Neurons and glial cells are both responsible for the production and formation of neuronal ECM. Upon the initial damage of CNS tissues by either traumatic injury or degenerative processes, the inflammatory responses in the CNS actively remodel the neuronal ECMs to prevent expansion of neuronal damage or to promote recovery of damaged tissue. ECM production and modifications of existing ECM molecules may either improve the recovery of neuronal damage or may aggravate inflammatory cycles, leading to chronic inflammation in the CNS [16].

The rationale behind the use of biocompatible hydrogels based on ECM components is that the active ingredients can reach the target cells/tissues with higher bioavailability due to immune system evasion or increased membrane permissibility. Most of the research pertains to the use of collagen as the main component, where it was found that injectable collagen quickly forms a gel to provide a continuous interface which may be useful in initial trauma response. However, as previously mentioned, collagen is not the main component of the ECM in CNS, and there is concern regarding its use as it may interact with integrins due to its adhesive properties [18]. Moreover, collagen has poor plasticity, degrades quickly, and lacks the mechanical properties necessary for support of regeneration [19]. One of the reasons why we did not perform collagen crosslinking is because it requires crosslinking agents such as DMTMM; however, this reaction with the tissue slices encapsulated in collagen would be difficult to perform. Such a strategy also limits mixing of the polymer chains, resulting in non-homogeneous crosslinking. Toxic effects of the reagents also cannot be ruled out, as we will not be able to purify the scaffold after crosslinking. On the other hand, HA appears to be an ideal candidate for hydrogel material selection. Moreover, it is highly water-soluble and non-immunogenic, although it is known to have poor adhesion [19,20]. It has previously been shown, however, that a crosslinked HA hydrogel alone was not sufficient to overcome the strong inflammatory response produced in the SCI rodent model [15].

### 4.2. HA-Based Hydrogels Differentially Promote Axonal Regeneration in DRGCs and SCSCs

DRGCs mostly consist of pseudomonopolar neurons that are sensory neurons without dendrites because the branched axon serves both functions. Therefore, DRGCs will give rise to a considerable number of sensory axons. On the other hand, SCSCs should give rise to a number of motor axons, apart from axons originating from sensory nerves and interneurons.

Axons from SCSCs grow much better on HA-based hydrogel compared to collagen regardless of the presence of neurotrophic factors. Axonal outgrowth from SCSCs is stimulated by the crosslinked HA-based hydrogel at the early timepoint; however, it is declined after an 8-day incubation. However, SCSCs axons continue to grow after 8 days when maintained in non-crosslinked Healon 5^®^ hydrogel. This can partially be explained by the mechanical and physicochemical properties of the non-crosslinked hydrogel. The latter is superior to crosslinked hydrogel at the late timepoint of 8 days regardless of the presence of neurotrophic factors, suggesting that different types of axons in SCSCs respond differentially to mechanical signals coming from different type of biomaterials and are not susceptible to the presence of neurotrophic factors.

Neurite outgrowth from DRGCs in our experiments was apparent from the second day in vitro, where it was accurately detected and characterized by the NeuriteSegmentation program. An increase in outgrowth on the fifth and eighth day in vitro suggested that the HA-based hydrogel, particularly in combination with neuroprotective factors, promotes axonal regeneration compared to collagen. When taking a closer look, we observe that sensory neurons from DRGCs regenerate at a much higher rate when maintained in a non-crosslinked hydrogel in the presence of neurotrophic factors, as compared to crosslinked hydrogel. As it was observed for axons derived from SCSCs, they sprouted to a similar extent in both crosslinked and non-crosslinked HA-based hydrogels after 5 days, while they deteriorated after 8 days in the crosslinked hydrogel. However, in the case of regeneration from DRGCs and in contrast to SCSCs, we observed a clear effect of the presence of neurotrophic factors. Therefore, it is reasonable to assume that sensory axons from DRGCs respond to a combination of signals that involve both neurotrophic factors and mechanical signals.

The process of crosslinking makes the hydrogel stiffer, thicker and more difficult for the body’s enzymes to break down and disintegrate compared to a non-crosslinked gel, which in our case is practically a liquid with high viscosity. It is reasonable to assume that some amount of neurotrophic factors would be trapped in the crosslinked hydrogel, thus not reaching the cultures in a sufficient concentration. This in combination with the stiffness of the gel could explain the deterioration observed after 8 days. It would also be a possible explanation for the presence of longer axons in cultures maintained in the non-crosslinked Healon 5^®^ in the absence of trophic factors.

Our results suggest though that the physical properties of both HA-based hydrogels differentially enhance axonal sprouting in SCSCs and DRGCs. These properties should be taken into consideration when designing future nerve grafts since peripheral nerves contain a variety of different type of axons. Moreover, nerve regeneration is a procedure that takes a considerable time period to reach the target, and therefore a stable biomaterial should be considered as an option.

### 4.3. Neuroprotective Effects of HA-Based Hydrogels

The aim of the qPCR experiment was to provide a quantifiable result to determine the extent of neuroprotection that the crosslinked HA hydrogel exerts on the NMDA-lesioned cultures. The potential neuroprotective effects of both crosslinked and non-crosslinked HA hydrogels were investigated on NMDA-lesioned neural tissue by gene expression analysis. The upregulated NeuN in all groups maintained in HA-based hydrogels suggest that the HA-based hydrogels, both crosslinked and non-crosslinked, possess neuroprotective effects. The Ct values of NeuN were normalized by using the values of actin-β. The rationale behind using a housekeeping gene is that its expression remains unchanged in both treatment and control groups. Actin-β is commonly used for this purpose and is one of the most common housekeeping genes, along with GAPDH [21].

Biocompatible hydrogels based on ECM components constitute a promising and novel research area of treatment for SCI injury. The ECM sustains cellular signalling mechanisms and is critical for functional and biochemical support [5,6]. In addition, a hydrogel, as a three-dimensional hydrophilic network, is capable of absorbing large volumes of water or biological agents, rendering them ideal candidates for drug delivery vectors and carriers or matrices for cells in tissue engineering [12].

HA-based hydrogels have been shown to have protective effects that can relieve damage by settling around or between the lesion like a protective membrane. We have been able to demonstrate a protective effect of HA-based hydrogels in our study. Therefore, apart from the axonal regenerative potential showed in the time-lapse experiment, the effect of our hydrogels as neuroprotective agents is also relevant in the acute stage immediately after spinal cord or avulsive nerve root injury.

Traumatic brachial nerve root avulsion is a devastating injury that, in the case presented, produced permanent paralysis of the extremities. Though nerve grafts are a promising route of treatment of avulsive injury, additional research into the biomechanics of nerve grafting and improvements in surgical technique are required [22]. Severe motoneuron death and inefficient axon regeneration often result in devastating motor dysfunction. Newly formed axons need to extend through inhibitory scar tissue at the CNS-PNS transitional zone before entering into a pro-regenerative peripheral nerve trajectory [23]. This could possibly be a direction of future research where neuroprotective properties of HA-based hydrogels could be beneficial in the treatment of avulsive nerve root injuries.

IL-1β in combination with other cytokines induces the expression of other interleukins, which suggests that IL-1β is involved in the modulation of autoimmune inflammation. It has been shown that in secondary damage after SCI, activated microglial cells release neurotoxic agents, where IL-1β is one of the most prominent mediators of neurotoxicity [24,25], and it has been suggested to play a regulatory role during neuroinflammation by promoting the expression of other proinflammatory mediators such as TNF-α and cyclooxygenase-2 [26]. Concentrations of IL-1β reach a peak at 1 h post-SCI, where mRNA can be detected 15 min after onset [27]. IL-1β is partly responsible for the upregulation of inducible NO-synthase (iNOS), which results in the synthesis of NO by neurons and microglia. In our experiment, we observed upregulation of IL-1β gene expression in the groups maintained on HA-based hydrogels compared to the NMDA group, which was not expected. That could probably be explained by the choice of method and timing in this experiment, because the gene expression analysis was performed 8 h after the onset of the experiments. In a previous analysis, we observed that protein levels of IL-1β start to deteriorate just four hours after the onset [25], suggesting that observation after 8 h could be too late in the NMDA-injured group, where the number of existing microglial cells that produce IL-1β is probably too small. The timing of observation may simply have been too late since the NMDA-injured cultures could already be severely degenerated.

## 5. Conclusions

In conclusion, we demonstrate that HA-based hydrogels in combination with neurotrophic factors are predisposed to neural protection and regeneration of axons from DRGCs and SCSCs. In future studies related to developing a nerve graft, we must consider that the type of biomaterial and/or neurotrophic factors selectively affect axonal outgrowth from different types of neurons. Our future plans involve the biomedical application of HA-based hydrogels and future development of nerve grafts, a procedure that needs to be verified through in vivo experiments. Our results indicate that HA-based hydrogels may be further designed and tailor-made according to the nature of the injured nerve or spinal cord region because motor and sensory axons respond differentially to different biomaterials. The limitations of this study concern the nature of in vitro models that cannot capture the inherent complexity of organ systems and the internal environment of the human body. Due to lack of blood supply, important interactions between sprouting axons and immune cells cannot be studied. However, our in vitro organotypic tissue cultures to some extent represent a microenvironment of cells and ECM that can still give valuable insight into how different biomaterials affect neural regeneration. Our next step in this direction will be to produce customized in vivo models wherein we expect to find similar results as in our in vitro models.

## Figures and Tables

**Figure 1 biomimetics-09-00140-f001:**
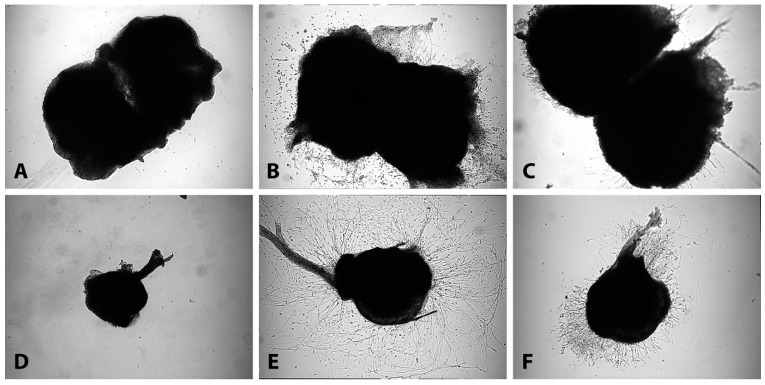
Axonal outgrowth from SCSCs and DRGCs at 5 days. Axonal outgrowth from SCSCs and DRGCs was observed in both crosslinked and non-crosslinked HA-based hydrogels while outgrowth in collagen gel was scarce((**A**,**D**): collagen, (**B**,**E**): non-crosslinked, (**C**,**F**): crosslinked). Cell migration was evident in cultures maintained in the non-cross-linked HA-based hydrogel.

**Figure 2 biomimetics-09-00140-f002:**
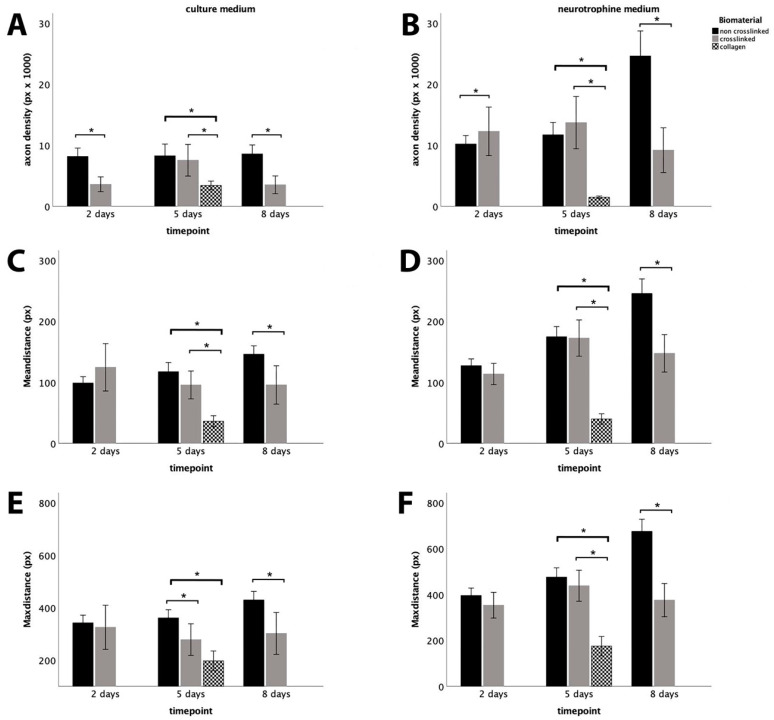
Quantitative analysis of axonal outgrowth from SCSCs using automated analysis with NeuriteSegmentation. Images (**A**,**C**,**E**) show the time lapse development of outgrowth parameters in SCSCs in the absence of NM. Images (**B**,**D**,**F**) show the respective parameters in the presence of NM. (* indicates signifigance *p* ≤ 0.05).

**Figure 3 biomimetics-09-00140-f003:**
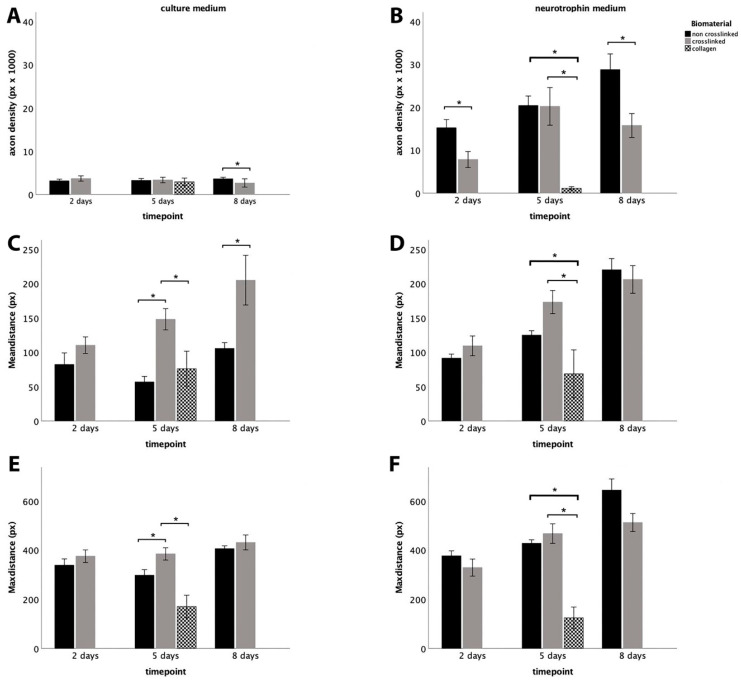
Quantitative analysis of axonal outgrowth from DRGCs using automated analysis with NeuriteSegmentation. Images (**A**,**C**,**E**) show the time lapse development of outgrowth parameters in DRGCs in the absence of NM. Images (**B**,**D**,**F**) show the respective parameters in the presence of NM. (* indicates signifigance *p* ≤ 0.05).

**Figure 4 biomimetics-09-00140-f004:**
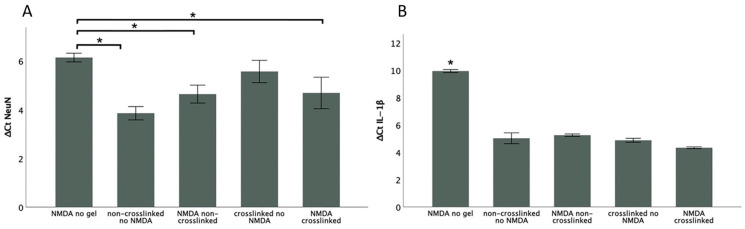
Gene expression analysis of NeuN and IL-1β from SCSCs using the ΔCt method. (* indicates signifigance *p* ≤ 0.05).

**Table 1 biomimetics-09-00140-t001:** Experimental groups in neuroprotection experiment.

Group Name	NMDA Injury	Type of Gel
NMDA no gel	Yes	No gel
Non-crosslinked HA no NMDA	No	Healon 5^®^
NMDA non-crosslinked HA	Yes	Healon 5^®^
Crosslinked HA no NMDA	No	Crosslinked HA-based
NMDA crosslinked HA	Yes	Crosslinked HA-based

**Table 2 biomimetics-09-00140-t002:** Taqman probes for the primers used in gene expression quantification.

Acronym	Name	Taqman Assay Number ID
ACTB	Actin-β	Mm02619580_g1
IL-1b	Interleukin 1 beta	Mm00434228_m1
Rbfox3/NeuN	RNA binding protein, fox-1 homolog 3	Mm01248771_m1

## Data Availability

The data presented in this study are available on request from the corresponding author.

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
