# Peer review of "Physically and Chemically Crosslinked Hyaluronic Acid-Based Hydrogels Differentially Promote Axonal Outgrowth from Neural Tissue Cultures"

_biomimetics, 2024, doi:10.3390/biomimetics9030140_

Round 1

Reviewer 1 Report

Comments and Suggestions for Authors

*Lines 24-25, don't think such a comment, made in the last sentence, should be in the abstract. It would rather be in the conclusion section.

*lines 29-33: Missing references.

* collagen was applied without crosslinking? Do the authors expect that there would be differences in better cell responses? I don't believe the authors should perform more experiments, mas I think it would be interesting to mention the reason they did not explore such alternatives based on early results that can be found in the literature.

*lines 215-217 "In the absence of trophic factors (CM) we observed longer axons in the non-crosslinked Healon 5® hydro-216 gel, however, there were no differences in terms of average axon length and axonal density between the gels (Figure 2C)" - There exists, latter on the paper, an explanation as to the slow migration of factors in the NM. In the case of CM, there is any explanation based on the physicochemical properties? Given the found results, would it be interesting to invest in chemical modification of HA aiming for chemical crosslinking? It is interesting to further discuss it.

* Result item titles are written quite weirdly, with conclusion parts being part of it (as can be seen in item 3.2).

*conclusions seem quite modest and could better explore the results

Comments on the Quality of English Language

The overall language style is appropriate em very clear for the reader.

Author Response

In respond to reviewer 1:

Thank you for your constructive comments. The manuscript has been meticulously revised according to the reviewers suggestions and the language has been edited by a native speaker.

Changes are indicated with blue in the manuscript.

  1. The comment in the last sentence of the abstract has been removed as the reviewer suggested.
  2. References have been added as the reviewer suggested, lines 30-32 in the revised version.
  3. Thank you for the comment about collagen.  The focus was HA-based hydrogel from the beginning. Based on previous experiments from our lab, we observed a satisfactory neuron preservation on collagen (Schizas 2014) however not that promising in terms of axonal regeneration. One of the reasons why we did not perform collagen crosslinking is because we need to add crosslinking agents such as DMTMM, however this reaction with the tissue slices encapsulated in collagen will be difficult to perform. Such a strategy also limits mixing of the polymer chains, resulting in in-homogeneous crosslinking. Toxic effects of the reagents also cannot be ruled out as we will not be able to purify the scaffold after crosslinking. A relevant paragraph has been added in the text lines 305-309.
  4. The presence of factors in NM certainly will have an impact on cell migration and axon length. We attempted both crosslinking and non-crosslinked gel for these experiments. We believe molecular interactions between HA chains and factors in NM will remain the same in both crosslinked and non-crosslinked hydrogels and therefore we did not observe any differences in average axon length and axonal density between the gels. A possible explanation of the presence of longer axons in cultures maintained in non-crosslinked gel in the absence of NM would be the physicochemical properties of the crosslinked gel, and more specific the stiffness. An explanation has been added in the text lines 349-351.
  5. The result items have been revised accordingly.
  6. The conclusion part has been revised according to the suggestion of the reviewer.

Reviewer 2 Report

Comments and Suggestions for Authors

The authors hypothesized that HA (rather than collagen)-based biomaterials could promote axon sprouting depending on the type of tissue culture, and placed spinal cord slice cultures (SCSCs) and dorsal root ganglion cultures (DRGCs) in collagen gels, physically crosslinked hyaluronic acid (HA)-based hydrogels (Healon 5®), and novel chemically crosslinked hyaluronic acid (HA)-based hydrogels, respectively. gels were cultured and the experimental results showed that different types of axons responded differently to different types of biomaterials. This is an interesting attempt, but major revisions are needed before publication.

1. To facilitate the reader's understanding, the authors are advised to improve the writing quality of the manuscript.

2. In the introduction section, the description of "Biocompatible biomaterials for axonal regeneration and neuroprotection" is too insufficient.

3. In the "Introduction" section, the authors should fully reflect the current status of the study, key issues to be solved and innovations.

4. In "Preparation and maintenance of Dorsal Root Ganglion Cultures:", the authors should indicate the mouse strain.

5. When the authors used real-time quantitative polymerase chain reaction (RT-qPCR) to determine the expression of IL-1β and NeuN, it is suggested that the authors should add the primer sequences used.

6. In the conclusion section, the authors should briefly summarize the study. In addition, this chapter should adequately address the limitations of this study, which is lacking in this manuscript.

Comments on the Quality of English Language

 Moderate editing of English language required

Author Response

In respond to reviewer 2:

  1. Thank you for your constructive comments. The manuscript has been meticulously revised according to the reviewers suggestions and the language has been edited by a native speaker.
  2. The term "Biocompatible biomaterials" in line 48 has been replaced by "Biocompatible ECM-based hydrogels"
  3. The introduction has been revised accordingly. Current status, key issues and innovations are summarised in lines 61-79. Please see also answer to reviewer 4 comment 5
  4. The mouse strain has been added in the text as suggested
  5. Thank you for your comment.  The primer sequence is not available in TaqMan probes because of a copyright agreement. The catalogue number of each probe is presented in table 2.
  6. The conclusion section has been revised according to the suggestions of the reviewer.

Reviewer 3 Report

Comments and Suggestions for Authors

The MS by Bajic et al. is devoted to influence of collagen or hyaluronic (HA) hydrogels on axonal outgrowth. The authors compared the effect of collagen, cross-linked and non-cross-linked hydrogels, as well as the ones with neuroprotectors on axonal outgrowth. It was found that non-crosslinked HA-based hydrogel with neuroprotector promotes the best effect axonal outgrowth.

The authors state that cross-linked HA-hydrogels are stiff and rigid and makes axonal outgrowth difficult after 8 days of observing. To obtain cross-linked hydrogel, the HA with about 10 % of modification CYS and CHO were used. Did the authors research hydrogels with less cross-linking degree? Why modified Has with 10 % of modification were chosen?

Also, Conclusion is insufficient. This section should be enhanced and presented main research results.

Author Response

In respond to reviewer 3:

Thank you for your constructive comments. The manuscript has been meticulously revised according to the reviewers suggestions and the language has been edited by a native speaker.

Changes are indicated with blue in the manuscript.

  1. The chemically crosslinked gels were designed to efficiently encapsulate tissue slices within the gel that can remain intact for a long period of time. For this purpose, we chose thiazolidine chemistry that demonstrate fast reaction kinetics. We chose 10% chemical modifications as below 10%, the gels are too soft and disintegrate in culture medium. Reducing crosslinking density also limit proper encapsulation of the tissue as 150 kDa HA used for making these gels are too small to provide additional viscoelasticity to the materials.
  2. The conclusion section has been revised according to the suggestions of the reviewer.

Reviewer 4 Report

Comments and Suggestions for Authors

The manuscript "Physically and chemically crosslinked hyaluronic acid-based hydrogels differentially promote axonal outgrowth from tissue cultures of spinal cord slices and dorsal root ganglion" is interesting and suitable for Biomimetics. The authors need to make some changes and clarifications:

1. There are too many words in the title.

2. The abstract does not seem convincing, please rewrite it. Please do not mention "collagen", it is not essential.

3. The article contains very few references, please add works for the chosen topic.

4. Line 78-79: "not collagen" is mentioned again.

5. Please argue the novelty and purpose of the work.

6. In Table 1, it would be more correct to write "crosslinked HA".

7. The conclusions do not provide a clear summary of the main points.

Comments on the Quality of English Language

Minor editing of the English language required

Author Response

In respond to reviewer 4:

Thank you for your constructive comments. The manuscript has been meticulously revised according to the reviewers suggestions and the language has been edited by a native speaker.

Changes are indicated with blue in the manuscript.

  1. Thank you for your comment. The title has been shortened according to the reviewers suggestion
  2. The abstract has been rewritten and "collagen" has been removed from the aims of the study. "Collagen" is though, mentioned as part of the groups.
  3. Additional references have been aded in the manuscript.
  4. The part of the text has been revised and "not collagen" is removed
  5. Thank you for your comment. We have revised the introduction (lines 61-79) so that the novelty of the work is highlighted.
  6. The table has been revised accordingly
  7. The conclusion part has been revised so that it summarises the main points. Limitations of the study are also mentioned in this part.

Round 2

Reviewer 3 Report

Comments and Suggestions for Authors

The authors made required corrections, so the MS can be accrpted for the publication in the current form

Reviewer 4 Report

Comments and Suggestions for Authors

The work was revised and improved according to the requirements.